# Heart Rate Influence on the QT Variability Risk Factors

**DOI:** 10.3390/diagnostics10121096

**Published:** 2020-12-16

**Authors:** Irena Andršová, Katerina Hnatkova, Martina Šišáková, Ondřej Toman, Peter Smetana, Katharina M. Huster, Petra Barthel, Tomáš Novotný, Georg Schmidt, Marek Malik

**Affiliations:** 1Department of Internal Medicine and Cardiology, University Hospital Brno, Faculty of Medicine, Masaryk University, Jihlavská 20, 625 00 Brno, Czech Republic; Andrsova.Irena@fnbrno.cz (I.A.); sisakova.martina@fnbrno.cz (M.Š.); Toman.Ondrej@fnbrno.cz (O.T.); Novotny.Tomas3@fnbrno.cz (T.N.); 2National Heart and Lung Institute, Imperial College, 72 Du Cane Road, Shepherd’s Bush, London W12 0NN, UK; K.Hnatkova@imperial.ac.uk; 3Wilhelminenspital der Stadt Wien, Montleartstraße 37, 1160 Vienna, Austria; psmetana@hotmail.com; 4Klinikum rechts der Isar, Technische Universität München, Ismaninger Straße 22, D-81675 Munich, Germany; Katharina.Huster@mri.tum.de (K.M.H.); barthel@tum.de (P.B.); gschmidt@tum.de (G.S.); 5Department of Internal Medicine and Cardiology, Faculty of Medicine, Masaryk University, Jihlavská 20, 625 00 Brno, Czech Republic

**Keywords:** QT variability, RR variability, QT variability index, underlying heart rate, sequential analysis of regression variance

## Abstract

QT interval variability, mostly expressed by QT variability index (QTVi), has repeatedly been used in risk diagnostics. Physiologic correlates of QT variability expressions have been little researched especially when measured in short 10-second electrocardiograms (ECGs). This study investigated different QT variability indices, including QTVi and the standard deviation of QT interval durations (SDQT) in 657,287 10-second ECGs recorded in 523 healthy subjects (259 females). The indices were related to the underlying heart rate and to the 10-second standard deviation of RR intervals (SDRR). The analyses showed that both QTVi and SDQT (as well as other QT variability indices) were highly statistically significantly (*p* < 0.00001) influenced by heart rate and that QTVi showed poor intra-subject reproducibility (coefficient of variance approaching 200%). Furthermore, sequential analysis of regression variance showed that SDQT was more strongly related to the underlying heart rate than to SDRR, and that QTVi was influenced by the underlying heart rate and SDRR more strongly than by SDQT (*p* < 0.00001 for these comparisons of regression dependency). The study concludes that instead of QTVi, simpler expressions of QT interval variability, such as SDQT, appear preferable for future applications especially if multivariable combination with the underlying heart rate is used.

## 1. Introduction

Despite all the recent technological and biochemical advances, evaluation of a standard 12-lead electrocardiogram (ECG) remains an essential diagnostic procedure. Among other aspects, the importance of ECG-based diagnostic tools is evident in population wide screening programmes in which the ease and low cost of ECG acquisition offers many practical advantages compared to more innovative investigations [1,2,3]. Naturally, electrocardiography also benefits from technological progress and different signal processing methods are being developed to assist ECG-based diagnoses well beyond the conventional visual interpretation [4,5,6].

One of these ECG processing technologies is based on the temporal measurement of beat-to-beat QT interval variability. The recent position statement by the European Heart Rhythm Association has shown that increased QT interval variability appears to be a risk marker of arrhythmic and cardiovascular death [7]. Indeed, the risk diagnostic value of QT interval variability has been reported in patients with cardiomyopathy [8,9], in long QT syndrome patients [10,11], in recipients of automatic implantable cardioverter defibrillators [12,13] as well as in a variety of other clinically and pathologically defined conditions [14,15,16].

In a number of previous studies, longer ECG recordings have been used [7]. Nevertheless, such recordings are not entirely realistic for wide screening purposes or indeed for day-to-day clinical practice. This is well recognised [17] and studies using ECG of standard 10-second duration for the investigation of QT variability have been reported including a proposal of normal physiologic values [18,19].

Lesser attention has been paid to the physiologic intra-subject variability of 10-second QT variability and to the physiologic correlates of different indices used to express short-term QT variability although reports have been published of poorer reproducibility of 10-second QT variability in comparison to the stability of other short-term ECG indices [18]. In particular, it has not been systematically researched whether the underlying heart rate and/or the underlying short-term variability of beat-to-beat intervals influences the QT variability and whether, in these respects, any noticeable differences exist between different measurement metrics that have previously been proposed to express the QT interval variability. The lack of understanding of heart rate influence on QT beat-to-beat variability contrasts the overwhelming knowledge on heart rate influence on QT interval duration. While corrections of QT interval duration for the underlying heart rate have now existed for a century [20,21], no such methodology exists for beat-to-beat QT variability.

Since such physiologic correlates of QT interval variability may influence the diagnostic and risk-prediction potency of the measurements and, since such correlates might need to be considered in future clinical studies, we have conducted a study investigating the rhythm-related correlates and reproducibility of 10-second QT variability in a large dataset of ECG recordings obtained during clinical pharmacology studies of healthy volunteers.

## 2. Methods

### 2.1. Investigated Population and Electrocardiographic Recordings

Clinical pharmacology studies conducted at 3 different locations enrolled 523 healthy volunteers including 259 females, with no statistical age differences between females and males (33.4 ± 9.1 years vs. 33.7 ± 7.8 years). Before study enrolment, all the volunteers had a normal standard clinical ECG and normal clinical investigation. Standard inclusion and exclusion criteria mandated for Phase I pharmacology studies [22] were used including negative recreational substances tests and negative pregnancy tests for females. The populations of the studies were based on standard calls for participation at pharmacology studies; no requirements on physical fitness and/or athletic training were made. All the source studies were ethically approved by the institutional ethics bodies (Parexel in Baltimore; California Clinical Trials in Glendale; and Spaulding in Milwaukee) and all subjects gave informed written consent to study participation and to scientific investigation of data collected during the studies.

In all volunteers, repeated long-term 12-lead Holter ECG recordings with Mason–Likar electrode positions were obtained covering the full day-time periods during which the subjects were not allowed to smoke and/or consume alcohol or caffeinated drinks. Those Holter recordings that were collected during days when the subjects were on no medication were analysed in the present study. The protocols of the different studies were also mutually consistent in respect of the clinical conduct during the drug-free baseline days. Since only drug-free data were used in the present investigation, further details of the source pharmacology studies are of no relevance.

Using previously described methods [23,24], multiple 10-second ECG segments were extracted from the long-term ECGs. The segments were selected with the aim of capturing different heart rates available in the Holter recording. That is, in addition to ECG segments obtained during protocol pre-specified study time-points, the complete day-time Holter recordings were scanned to obtain heart rates of all measurable 10-second extractions. From these, ECG segments were selected so that in each recording, the complete range of heart rates was uniformly covered [24]. All the extracted 10-second segments contained only sinus rhythm recordings and were free of any ectopic beats.

In each of these ECG segments, the QT interval was measured following published procedures [23,24] that included repeated visual controls of all the measurements and assurance that corresponding ECG morphologies were interpreted in a consistent way [25]. The visually verified QT interval measurements were made in the representative median waveforms of the 10-second segments (sampled at 1000 Hz) with the superimposition of all 12 leads on the same isoelectric axis [26,27]. In more detail: The QRS onset and T wave offset points were initially generated by validated signal processing algorithms applied to each extracted 10-second ECG segment. Subsequently, these positions were projected on the superimposed representative waveforms and their positions were checked by two independently working cardiologists. These checks were made on computer screens with a display resolution of 1 millisecond per 1 pixel. Where necessary, the cardiologists used the graphics displays to correct the QRS onset and T wave offset positions manually. When the two cardiologists disagreed, a senior third cardiologist reconciled their differences. In this way, systematically consistent positions of the QRS onset and T wave offset were obtained for the representative waveform of each extracted ECG segment.

### 2.2. Beat-to-Beat QT Interval Measurements

Using a previously proposed technique [28,29], QT interval was projected to individual beats within the 10-second ECG by finding the maximum correlation between the representative waveform (in which the original measurement was made) and the signal of individual QRS-T complexes. The maximum correlations were identified separately for the surroundings of the QRS onset and of the T wave offset. Since it has previously been observed that this process might lead to slightly different results when applied to different ECG leads [30], the cross-correlation technique was applied to the vector magnitude of algebraically reconstructed orthogonal leads [31]. ECG segments were excluded from analysis if noise pollution prevented the beat-to-beat measurements of the QT interval to be made reliably.

### 2.3. QT Interval Variability Expressions

In each 10-second ECG, heart rate was measured in beats per minute (bpm) based on the average duration of all RR intervals. For the purposes of investigating the QT interval variability and its physiological correlates, a further 6 indices were obtained for each 10-second ECG:
Standard deviation of all RR intervals (SDRR),Coefficient of variance of RR intervals (RR_cvar_ = SDRR/$\bar{\mathrm{RR}}$, where $\bar{\mathrm{RR}}$ is the average of all RR intervals),Standard deviation of all QT intervals in individual beats (SDQT),Coefficient of variance of QT intervals (QT_cvar_ = SDQT/$\bar{\mathrm{QT}}$, where $\bar{\mathrm{QT}}$ is the average of all QT intervals),Proportion of QT and RR interval variances (QT_var_/RR_var_ = SDQT^2^/SDRR^2^),QT variability index (QTVi = ${\mathrm{QT}}_{\mathrm{cvar}}^{2}{/\mathrm{RR}}_{\mathrm{cvar}}^{2}$).

Of these indices, the QT variability index was the first index to be introduced [28] and is perhaps the most frequently used QT variability expression [7] although it includes not only QT interval but also the RR interval variability.

### 2.4. Data Investigations

To investigate the heart rate effects and the correlates of the different indices, the available data of the described indices were used in the following investigations.

#### 2.4.1. Effects of Heart Rate

To investigate the effect of heart rate in principle, averages of the 6 indices were obtained, in each study subject separately, for ECGs with heart rate between 50 and 75 bpm and for ECGs with heart rates between 75 and 100 bpm. Cumulative distributions of these averages were produced.

#### 2.4.2. Influence of Age

For each index, the intra-subject averages of the indices obtained for heart rate ranges of 50–75 and 75–100 bpm were related to the age of the subjects. Linear regressions were used to investigate the relationship.

#### 2.4.3. Intra-Subject Variability

In each study subject, standard deviations of the indices were also obtained for heart rate ranges of 50–75 and 75–100 bpm. These were used to obtain the intra-subject coefficient of variances of each index in these heart rate bands. Similar to the intra-subject means of the indices, the cumulative distributions of the intra-subject coefficient of variances were constructed.

#### 2.4.4. Intra-Subject and Inter-Subject Relationship between the Indices

Firstly, within the data of each subject separately, Spearman rank correlations were calculated between selected pairs of the indices. The cumulative distributions of the correlation coefficients were constructed.

Secondly, to investigate the proportional relationship between different indices (including the heart-rate relationship), sequential analysis of regression variance was used. That is, when investigating how a combination of indices A and B relates to index Z, we considered a multivariable linear regression Z = β_0_ + β_1_A + β_2_B + e_AB_, and compared the regression residuals e_AB_ of this regression model with the residuals of univariable regressions Z = ξ_0_ + ξ_1_A + e_A_ and Z = ζ_0_ + ζ_1_B + e_B_ (where β_i_, ξ_i_, and ζ_i_ are numerical coefficients obtained by solving the standard linear regression equations, and e_•_ are zero centred normally distributed residuals). These linear regressions were obtained for each subject separately. If, in the study population, the proportions (e_A_ − e_AB_)/e_A_ were smaller than the proportions (e_B_ − e_AB_)/e_B_, it was concluded that the index Z was influenced by index A more than by index B. This is because the proportion (e_A_ − e_AB_)/e_A_ shows how much of the residual e_A_ (i.e., a residual left after applying the regression of Z to A) can be explained by further regression to B. Where dictated by the definition of index Z, the values of indices A or B were replaced by their reciprocal values 1/A or 1/B in the regression equations.

Similar considerations were made when considering three predictor indices A, B, and C. Residuals e_ABC_ of linear regression Z = β_0_ + β_1_A + β_2_B + β_3_C+e_ABC_ were used as a reference in comparisons between (e_A_ − e_ABC_)/e_A_ and (e_B_ − e_ABC_)/e_B_, between (e_AB_ − e_ABC_)/e_AB_ and (e_AC_ − e_ABC_)/e_AC_, and likewise for further combinations of predictor indices.

### 2.5. Statistics and Data Presentation

Descriptive data are presented as means ± SD. Comparisons between females and males were based on a two-sample two-tail *t*-test assuming different variations between compared datasets. Within-subject comparisons (e.g., comparisons of the indices between the two heart rate bands, or comparisons between coefficients of variance of different indices) were based on a paired two-tail *t*-test. The significance of linear regression slopes between age and the investigated indices was tested using the Fisher–Snedecor F distribution. The comparisons between the proportions of regression residuals used non-parametric paired Wilcoxon test. The calculation of the multivariable linear regressions repeated in different study subjects utilised in-house matrix manipulation software package programmed in C++. Statistical tests used IBM SPSS package, version 27. *p* values below 0.05 were considered statistically significant. Because of interdependence between the different indices, no correction for multiplicity of statistical testing was made.

## 3. Results

The analyses of the study were based on the total of 657,287 individual 10-second ECG samples in which reliable beat-to-beat QT interval measurements were made (only 1.1% of the original data in which QT interval measurement was made and visually confirmed in the representative waveform had to be excluded because of problems with beat-to-beat measurements). On average, there were 1247 ± 221 and 1266 ± 218 ECG segments processed in female and male subjects (no significant differences between the sex groups). 

Figure 1 shows the cumulative distributions of the 6 indices measured in ECGs of the two heart rate bands. In all indices with the exception of 10-second RR interval coefficient of variance, the heart rate effect was highly statistically significant in both sexes (*p* < 0.00001 for all the comparisons). The difference of the RR_cvar_ values in the two heart rate bands was only significant in males (*p* < 0.00001) but was not significant in females. The only significant sex differences were larger SDRR and RR_cvar_ in females at heart rates 50–75 bpm (*p* = 0.02 and 0.002, respectively) and larger SDQT in females at both heart rate bands (*p* = 0.002 for 50–75 bpm, and *p* = 0.02 for 75–100 bpm).

All the indices, except for QT_cvar_ at heart rates of 75–100 bpm, were statistically significantly related to age (both in females and males, with *p* values ranging from 0.005 to <0.00001). While RR variability (both SDRR and RR_cvar_) decreased with age, the other QT variability indices increased with age. While generally, the increase of QT variability with advancing age was moderate, it was stronger for the data measured at 75–100 bpm compared to 50–75 bpm. Figure 2 shows scatter diagrams of the relationship with age for RR_cvar_, QT_cvar_, and QT variability index. From the practical point of view, a steep relationship to age was observed for the QT variability index measured at heart-rates of 75–100 bpm where it reached 0.020 and 0.015 increases of the index for each year of age in females and males, respectively. While the slopes of the SDQT related to age were statistically significant, they were rather shallow.

The distributions of intra-subject coefficients of variance of the different indices are shown in Figure 3. As seen in the figure, the measurements of SDQT and QT_cvar_ were, within individual subjects, more reproducible than the measurements of SDRR and RR_cvar_ which, in turn, were more reproducible than the QT_var_/RR_var_ ratio or the QT variability index. All these differences were highly statistically significant in both sexes (all *p* < 0.00001). In addition to these principal reproducibility results, we also observed that SDRR, RR_cvar_, QT_var_/RR_var_ ratio, and the QT variability index were less reproducible at the higher heart rates 75–100 bpm compared to the lower heart rates of 50–75 bpm (again, all *p* < 0.00001). At the slower heart rate of 50–75 bpm, the QT_var_/RR_var_ ratio, and QT variability index were also less reproducible in males compared to females (both *p* < 0.00001) but this difference was not present at the higher heart rates. Further differences shown in the distribution graphs of Figure 3 were occasionally also statistically significant but were numerically minimal and, thus, without obvious implications.

Figure 4 shows the distributions of intra-subject rank correlation coefficients between selected indices. Importantly, with the exception of RR_cvar_, all the indices were systematically related to heart rate. While the SDRR decreased with increasing rate, SDQT, QT_cvar_, QT_var_/RR_var_ ratio, and the QT variability index were all systematically increasing with increasing heart rate. As expected, QT_var_/RR_var_ ratio and QT variability index were also, within each subject, positively correlated with SDQR and negatively correlated with SDRR. Figure 5 shows scatter diagrams between intra-subject means of selected indices. (Note that while the data shown in Figure 5 are individual means—each subject is represented by one marker—the correlation coefficients summarised in Figure 4 were calculated within each subject separately). Again, as expected, a strong relationship between SDQT and the QT variability index is seen also at the population level—especially at the higher heart rates 75–100 bpm. Figure 5 should not be interpreted as a suggestion of “correctable” relationships (note the large spreads of the individual points). Rather, the figure demonstrates the differences in the strength of the relationships between different indices and the heart rate influence on these relationships.

The results of the sequential analysis of the regression variance are shown in Figure 6. The top panels of the figure show that within individual subjects, the SDRR and QT_cvar_ are more strongly influenced by heart rate than by the short-term RR variability represented by SDRR and RR_cvar_. In both cases, the relative regression residuals were much larger for SDRR than for heart rate (*p* < 0.00001 for both sexes in both cases). The same results with the same strong statistical significances were obtained when considering regressions QT_cvar_ = β_0_ + β_1_ × HR + β_2_ × SDRR and SDQT = b_0_ + b_1_ × HR + b_2_ × RR_cvar_. The left middle panel of the figure shows that when relating the QT_var_/RR_var_ ratio to both QT_cvar_ and the reciprocal of RR_cvar_, there was only a non-significant trend towards larger relative residuals left by QT_cvar_. The middle right panel shows that the QT variability index was more strongly influenced by the reciprocal of SDRR than by SDQT (*p* < 0.00001 in both sexes). Similar results were obtained when relating the QT variability index to a combination of heart rate, reciprocal of SDRR and SDQT (the bottom panels of Figure 6). Relative residuals left by the reciprocal of SDRR were lower than those left by SDQT and similarly, relative residuals left by a regression combination of heart rate and the reciprocal of SDRR were lower than those left by a combination of heart rate and SDQT (all the comparisons in these cases gave *p* < 0.00001 for both sexes). Note that the panels of Figure 6 also allow visual comparisons—when the majority of the points (each representing the relative residuals in the ECGs of one individual subject) appear above the line of identity, the influence of the predictor (or predictors) shown on the horizontal axis is stronger than the influence of the predictor(s) shown on the vertical axis.

## 4. Discussion

The study leads to three distinct observations that all appear to be of practical importance. Firstly, the study shows convincingly that in healthy subjects, the different indices used in the vast majority of studies reporting QT interval variability [7] are all strongly related to the underlying heart rate. Secondly, while the QT variability index is the most popular numerical expression of QT interval variability, we have also observed that it is substantially less reproducible compared to the simpler expressions such as the standard deviation of the QT intervals. Finally, we have also noted that among the investigated indices of QT variability, the QT variability index was most strongly related to age of healthy subjects. The reason for this age dependency is likely based on the age-related decline of RR variability.

Since its inception [28], it has been understood that QTVi is a combination of both QT and RR interval variability. The somewhat surprising aspect of our results is the proportion of the influences based on the sequential analysis of the regression variance. As shown in Figure 6, the RR interval variability is the dominant determinant of QTVi while the QT interval variability provides only a secondary influence. This needs to be considered together with our other finding that shows that the underlying heart rate rather than RR variability drives the QT variability measured by SDQT or QT_cvar_. These observations have clear implications for further utility and research of QTVi. Many studies have repeatedly shown the risk-prediction capability of QTVi [32,33,34,35,36]. Nevertheless, both increased heart rate [37,38] and decreased heart rate variability [39,40] are well recognised strong risk factors. Since, as we demonstrated, both increased heart rate and decreased RR variability increase the QTVi values, it is legitimate to ask to which extent the QTVi-based risk prediction is driven by true QT variability. While combinations of different factors into composite measures and/or scoring systems are valid methods for prospective diagnostic and risk studies, physiologic understanding and diagnostic utility of QT interval variability can be masked by its combination with other risk factors especially if these factors influence the combined values as strongly as we have shown. Still, if QTVi values are used in future prospective studies, the strong relationship to age needs to be considered; different diagnostic dichotomy limits are needs for different age groups.

Therefore, we are of the opinion that future investigative studies of QT variability would be better served by using simpler expressions of QT variability, such as the standard deviation of individual QT interval durations. This is also supported by our observation that SDQT showed substantially tighter intra-subject reproducibility compared to QTVi. (Note, however, that intra-subject coefficients of variance of around 50% to 60% as shown in Figure 3 for SDQT indicate still fairly variable results albeit much more stable compared to coefficient of variance around 200% that we observed for QTVi). Nevertheless, even with SDQT or QT_cvar_, the influence of heart rate still needs to be considered. Multivariable analyses involving the QT variability indices together with the underlying heart rate may be proposed.

QT variability assessment has already been implemented in commercial Holter systems [41]. It can, therefore, be advocated that future clinical risk-assessment studies combine the simpler QT variability indices with other risk-stratification techniques ranging from heart rate variability [42] and heart rate turbulence [43,44] to deceleration capacity [45] and T wave alternans [46]. As all these indices can be obtained from the same Holter recordings, their multivariable comparisons and combinations might be applied to a variety of clinically well-defined populations. Implementation of the simpler expressions of QT variability should also be possible in standard bedside models of ECG equipment (which already report the underlying rate).

There is little independent data available which we could use to validate our principal results, especially those obtained by the sequential analysis of the regression variance. Other observations made in the study appear to agree with previous publications. The observation of larger SDRR in females compared to males that we observed at slower but not faster heart rates is consistent with reports of sex differences in frequency components of heart rate variability during resting but not during sympathetically stimulating conditions [47]. The larger SDQT values in females are likely related to the sex difference in QT interval duration since there was comparatively lesser difference observed with QT_cvar_. Because of the substantial heart rate influence on the measured QT variability indices, we are also unable to compare our measurements with previously published normal values [19]. Ranges and distributions of the values that we have measured for the different indices are shown in Figure 1 including their changes due to the heart rate differences.

Since the QT variability index is predominantly influenced by beat-to-beat RR interval differences, its relationship to age most likely expresses the age-related decline of heart rate variability [42] rather than the age-associated changes of QT interval duration [48]. 

## 5. Limitations

Limitations of our study also need to be considered. While a number of previous QT variability studies used longer ECGs, we concentrated on 10-second ECG segments because these are more relevant for practical purposes. This also means that we are unable to comment on whether the very same observations would be obtained also with longer recordings. Nevertheless, since every longer recording is, in principle, a series of shorter segments, it is unlikely that with longer ECGs, our results would be very different. Although we have accepted beat-to-beat QT interval measurements only when a closed fit was found between the individual beat images and the representative median waveforms, some residual influence of ECG noise cannot be excluded entirely. Since ECG noise can be expected to increase with physical activity which, in turn, leads to increased heart rates, our observations of heart rate influence might have been overestimated. However, since positive intra-subject correlations between heart rate and the QT variability indices were found in practically every subject of the study (see Figure 4) any noise-related overestimating of the heart rate dependency might have only been marginal. The investigated population included neither very young nor very old subjects. The investigations of the relationship to age were, therefore, limited to the available age ranges. Finally, since the study data were obtained from clinical pharmacology investigations in healthy subjects, we are unable to comment on whether the same results would have been found if researching populations with clinically well-defined pathological characteristics. Nevertheless, our observations still have an impact on clinical studies in the same way as other physiologic characteristics influence clinical investigations and diagnoses. In particular, while increased QT variability has previously been reported in different studies of congenital long QT syndrome patients [7], we cannot comment on the heart rate influence of QT variability in these patients. Similarly, we cannot comment on whether multivariable QT variability and heart rate assessment could serve the diagnostics of acquired (e.g., drug-induced) long QT syndrome [49] and whether it might increase the power of relevant clinical studies [50].

## 6. Conclusions

Despite these limitations, the study shows that the underlying heart rate and the underlying RR interval variability are crucial determinants of the standard indices of QT interval variability. This influence is of particular importance for future applications of the popular QT variability index which is less influenced by true QT variability than it is by the heart rate and RR interval characteristics. The QT variability index also shows substantially low intra-subject reproducibility. Simpler expressions of QT interval variability such as the standard deviation of QT interval duration in individual beats might be preferable in future applications especially if multivariable combination with the underlying heart rate is used.

## Figures and Tables

**Figure 1 diagnostics-10-01096-f001:**
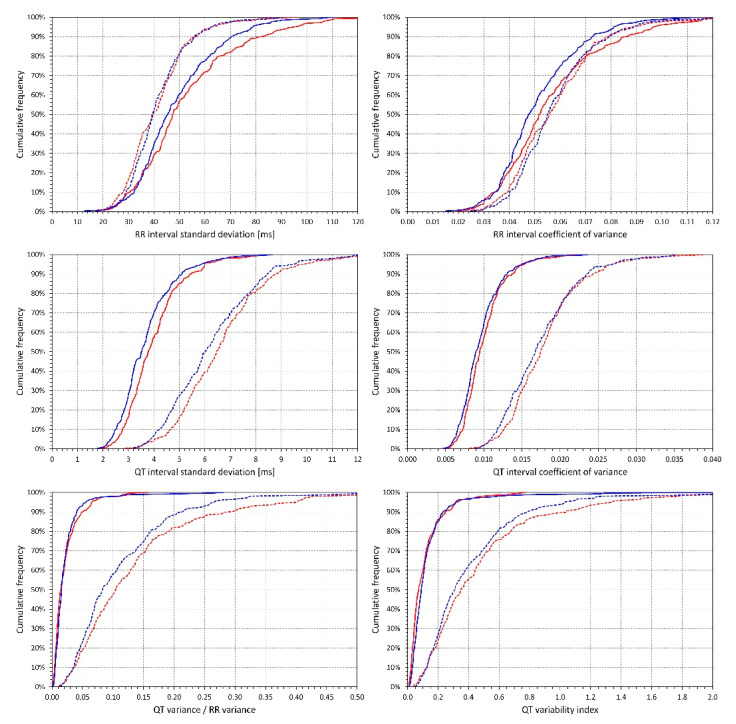
Values of measured indices. For each of the investigated indices (see the labels of the horizontal axes of individual panels) the figure shows the cumulative distributions of intra-subject means calculated over electrocardiograms (ECGs) with heart rate between 50 and 75 bpm (full lines) and between 75 and 100 bpm (dashed lines). The red and blue lines correspond to female and male subjects, respectively.

**Figure 2 diagnostics-10-01096-f002:**
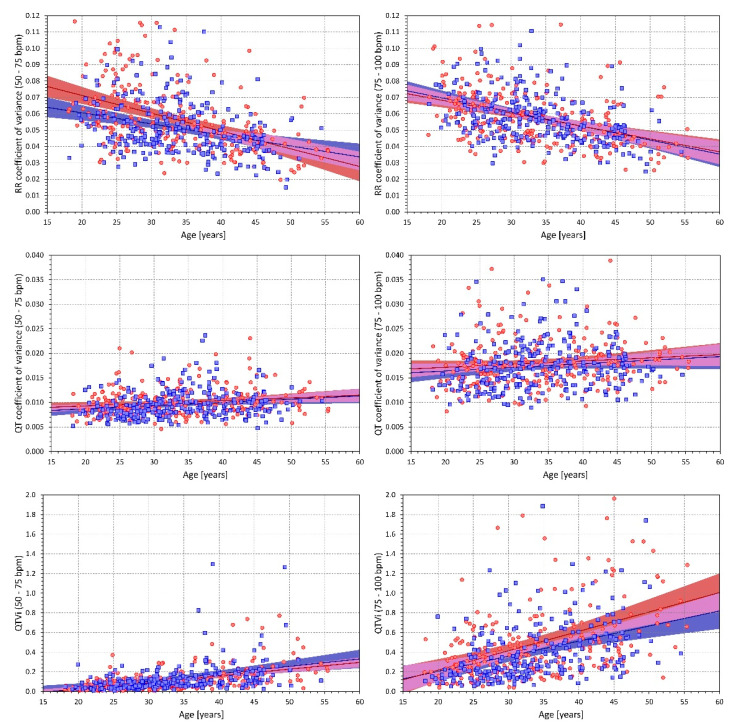
Relationship to age. Scatter diagrams of the relationship between age of the study subjects (horizontal axes) and RR interval coefficient of variance (top panels), QT coefficients of variance (middle panels), and QT variability index (bottom panels). The panels on the left and on the right show the relationship of age to intra-subject means calculated over ECG with heart rate between 50 and 75 bpm and between 75 and 100 bpm, respectively. In each panel, the red circles and blue squares correspond to female and male subjects, respectively. The solid red and solid blue lines show the linear regressions between the ages and the intra-subject mean values. The red shaded and blue shaded areas are the 95% confidence bands of the regression lines; the violet areas are the overlaps between the confidence bands of the sex-specific regressions.

**Figure 3 diagnostics-10-01096-f003:**
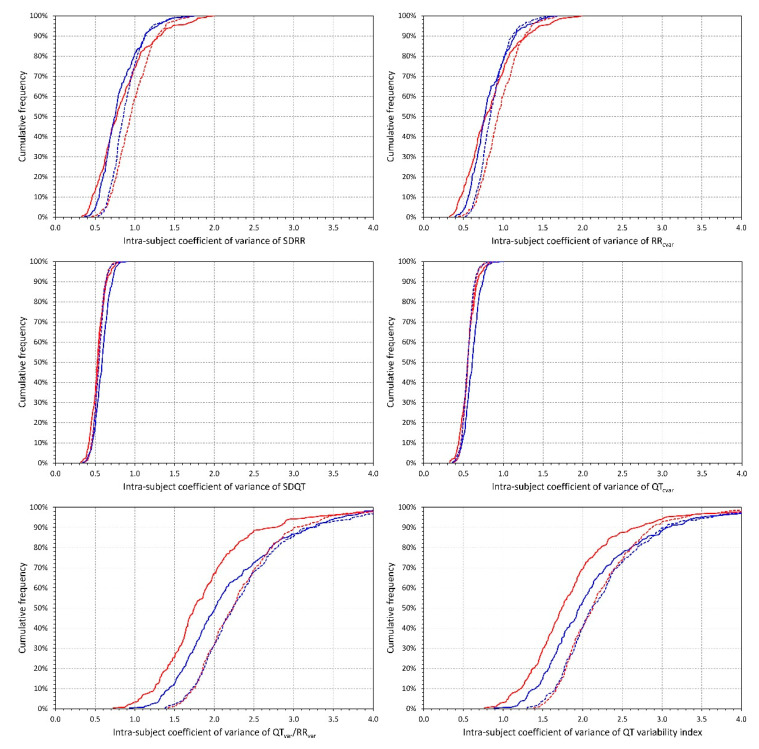
Intra-subject coefficients of variance. For each of the investigated indices (see the labels of the horizontal axes of individual panels) the figure shows the cumulative distributions of intra-subject coefficient of variance of the given index calculated over ECGs with heart rate between 50 and 75 bpm (full lines) and between 75 and 100 bpm (dashed lines). The red and blue lines correspond to female and male subjects, respectively.

**Figure 4 diagnostics-10-01096-f004:**
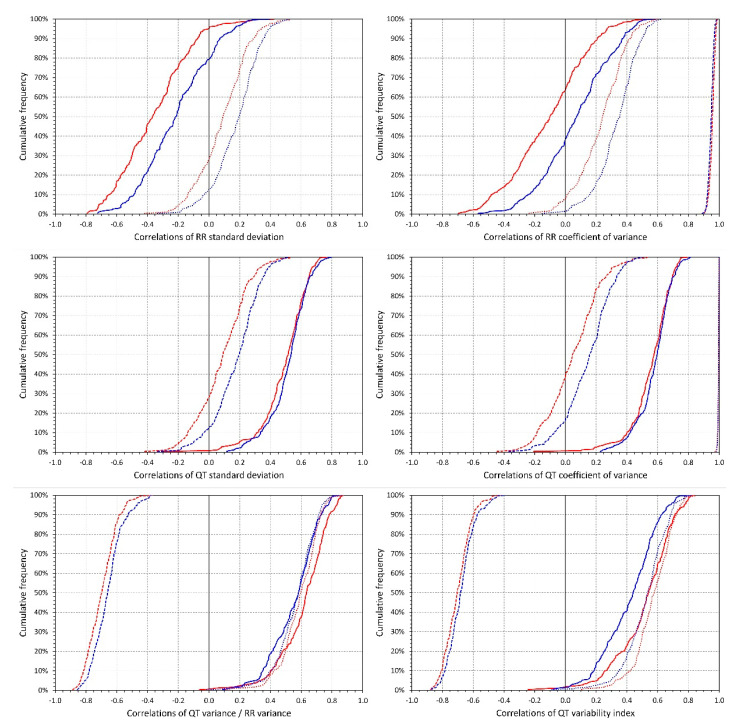
Intra-subject rank correlations. For each of the investigated indices (see the labels of the horizontal axes of individual panels) the figure shows the cumulative distributions of intra-subject Spearman rank correlation coefficients (calculated over all available ECGs in the given subject) between the given index and the underlying heart rate (full lines), standard deviation of RR intervals (dashed line), and standard deviation of QT intervals (dotted line). The red and blue lines correspond to female and male subjects, respectively.

**Figure 5 diagnostics-10-01096-f005:**
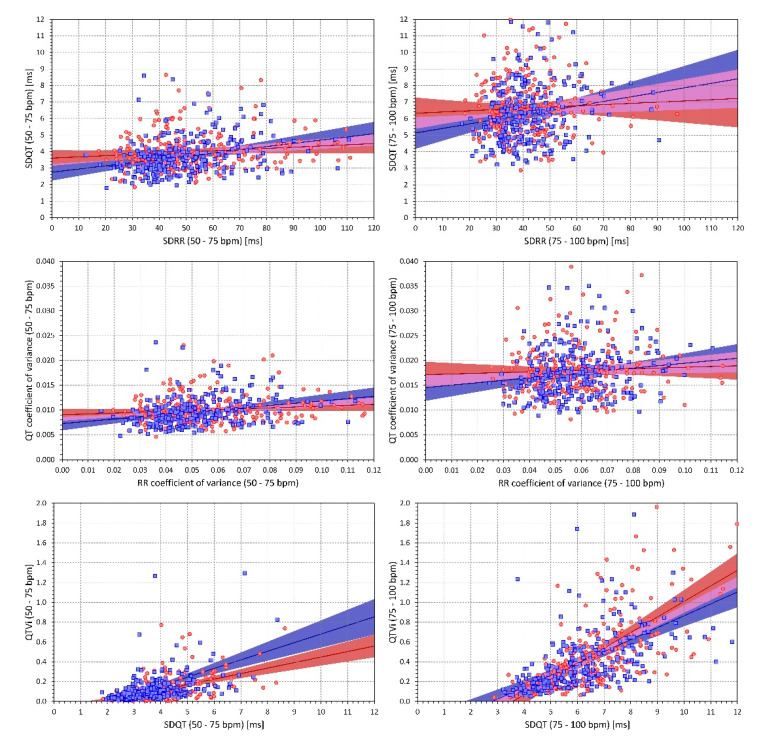
Inter-subject relationship of the measured indices. Scatter diagrams of the inter-subject relationships between intra-subject means of SDRR and SDQT (panels on the top), RR_cvar_ and QT_cvar_ (panels in the middle), and SDQT and QT variability index (panels at the bottom). Panels at the left and on the right show the indices calculated over ECGs with heart rates 50–75 bpm and 75–100 bpm, respectively. In each panel, the red circles and blue squares correspond to female and male subjects, respectively. The solid red and solid blue lines show the linear regressions between the intra-subject mean values of the compared indices. The red shaded and blue shaded areas are the 95% confidence bands of the regression lines; the violet areas are the overlaps between the confidence bands of the sex-specific regressions.

**Figure 6 diagnostics-10-01096-f006:**
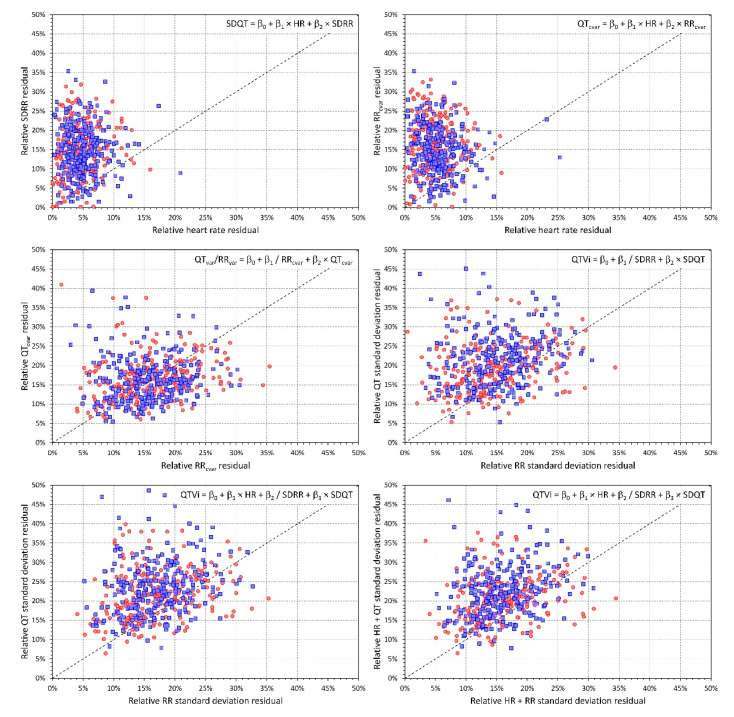
Sequential analysis of the regression variance. Results of the sequential analysis of the regression variance (see the text for details). Each panel corresponds to a given linear regression estimate (see the formulas at the top right of the panels) and shows the relative residuals, that is the proportions (e_A_ − e_AB_)/e_A_ or (e_AB_ − e_ABC_)/e_AB_ as explained in the text. Scatter diagrams in the separate panels show the relationship between relative residuals of predictors used in the multivariable regression. The labels of the axes have the form “Relative A residuals” or “Relative A+B residuals” meaning the proportions (e_A_ − e_AB_)/e_A_ or (e_AB_ − e_ABC_)/e_AB_. In each panel the dashed line shows the line of identity. In each panel, the red circles and blue squares correspond to female and male subjects, respectively (note that the multivariable linear regressions and their residuals regressions were evaluated in each subject separately using all the ECGs available for the given subject).

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
