# Peer review of "Heart Rate Influence on the QT Variability Risk Factors"

_diagnostics, 2020, doi:10.3390/diagnostics10121096_

Round 1
Reviewer 1 Report
I read with interest of an extensive study. The authors concludes that the study has following contribution:
Firstly, the study shows convincingly that in healthy subjects, the different indices used in the vast majority of studies reporting QT interval variability are all strongly related to the underlying heart rate.
-Comments: this is not a new conclusion, thus QTc is used for calculation to adjust to RR variability
Secondly, while the QT variability index is the most popular numerical expression of QT interval variability, we have also observed that it is substantially less reproducible compared to the simpler expressions such as the standard deviation of the QT intervals.
- Is this practically possible to report SD of the QT intervals in day-to-day practice
Finally, we have also noted that among the investigated indices of QT variability, QT variability index was most strongly related to age of healthy subjects. The reason for this age dependency is likely based on the age-related decline of RR variability.
- Again, this is a known fact.
The paper is very well written and the study is well done. The contribution of this paper is questionable for clinical practice.
Author Response
Reviewer #1
Firstly, the study shows convincingly that in healthy subjects, the different indices used in the vast majority of studies reporting QT interval variability are all strongly related to the underlying heartrate.
-Comments: this is not a new conclusion, thus QTc is used for calculation to adjust to RR variability
RESPONSE: This comment was likely based on misunderstanding. We are not talking about the QT interval duration but about the beat-to-beat QT interval variability. We have now expanded the Introduction section to make the distinction more visible. (Lines 61-64.)
Secondly, while the QT variability index is the most popular numerical expression of QT interval variability, we have also observed that it is substantially less reproducible compared to the simpler expressions such as the standard deviation of the QT intervals.
- Is this practically possible to report SD of the QT intervals in day-to-day practice
RESPONSE: There are algorithms available to assess short term beat-to-beat QT interval variability in 10-second electrocardiograms. This has already been shown in a number of publications – e.g. our references [19]. In the Discussion section, we have also addressed the point of implementation of these algorithms in the standard ECG equipment. (Lines 345-347.)
Finally, we have also noted that among the investigated indices of QT variability, QT variability index was most strongly related to age of healthy subjects. The reason for this age dependency is likely based on the age-related decline of RR variability.
- Again, this is a known fact.
RESPONSE: We again believe that this comment has likely been based on misunderstanding. While QTc duration is known to be age dependent, we are not talking about QTc interval duration but about the so-called QT variability index (as first published by Berger at al some 25 years ago – our reference [28] – and frequently followed in more recent studies). We have now added a discussion explaining that the QTc age dependency and the QTVi age dependency appear to be different processes. (Lines 359-361)
The paper is very well written and the study is well done. The contribution of this paper is questionable for clinical practice.
RESPONSE: The clinical applicability of QT interval variability has been, among others, addressed in our references [7-16]. This mentioned in the second paragraph of Introduction. (Lines 42-48)
Reviewer 2 Report
This paper presents an relevant problem in hospital and practice. On the one hand, HRV is an meaningful measure in cardiology, on the other hand, QT duration is a significant measure in ECG to estimate an increased risk for sudden cardiac death.
The presentation is very well prepared in terms of methodology, presentation of results and discussion.
Some comments should be clarified.
Were the subjects normal persons without regular training, athletes?
Were the ECG times measured at defined (rest or after activity) times?
How were they measured? By meansof ECG ruler (visual) or with an automatic evaluationby a PC program in the ECG device? Which company ?
One reference to the measurement of HR turbulence is missing (Schmidt et al., Lancet 1999), this also gives a good prediction of SD.Which approach is superior?
Were the results also tested in patients with LQT syndrome ?
The point clouds in Fig.5 and 6 should be explained additionally for the reader, as they do not reliably provide a correlation calculation in this kind of cluster.
Are the algorithms available on standard ECG devices for additional evaluation ?
Are there reference values for the normal findings ?
Author Response
Reviewer #2
Were the subjects normal persons without regular training, athletes?
RESPONSE: We have now explained that the study population was based on standard call for clinical pharmacology investigation. No requirement for physical fitness and/or training was made. (Lines 77-79)
Were the ECG times measured at defined (rest or after activity) times?
RESPONSE: The amended text now also explains that the ECG were selected (a) during pre-defined study time-points and (b) during free scans that aimed at capturing different heart rates. (Lines 92-95)
How were they measured? By means of ECG ruler (visual) or with an automatic evaluation by a PC program in the ECG device? Which company ?
RESPONSE: We have now expanded the text on the measurement considerably. The measurement of the QT interval was, in each ECG segment, verified visually with manual correction where appropriate. The measurement of the best-to-beat variability of the QT interval was based on auto-correlation estimates that were made automatically (the algorithm used for the QT variability measurements was fully validated). (Lines 103-111)
One reference to the measurement of HR turbulence is missing (Schmidt et al., Lancet 1999), this also gives a good prediction of SD. Which approach is superior?
RESPONSE: We know the principles of HR turbulence well (indeed, the senior author of this article was also the second author of the Lancet 1999 paper as well as the corresponding author of the ISHNE standards of HR turbulence published in JACC 2008). Nevertheless, since we have, by study design, used only ECG segments without any ventricular premature beats, the principles of HR turbulence are independent of we are describing. We have included a brief discussion of HR turbulence as well as of other risk prediction phenomena in the discussion. There, we also address the comment on future comparisons of risk prediction capabilities. The present data do not allow us to make such a comparison directly. (Lines 340-347)
Were the results also tested in patients with LQT syndrome ?
RESPONSE: QT variability have repeatedly been studies in congenital LQTS patients. Our data do not allow us to answer the obvious questions whether the same rate relationship exists in congenital LQTS and whether incorporating rate relationship would make the phenomenon of QT variability useful in the assessment of acquired (e.g. drug-induced) LQTS. This all have now been included in the discussion of the article. (Lines 381-386)
The point clouds in Fig.5 and 6 should be explained additionally for the reader, as they do not reliably provide a correlation calculation in this kind of cluster.
RESPONSE: The meaning of Figures 5 and 6 is rather different. Figure 5 shows the relationships between the measurements whilst Figure 6 shows comparisons of regression residuals. We have now expanded the explanations of both figures to explain the meaning of the figures in more detail. (Lines 253-255 and 287-291)
Are the algorithms available on standard ECG devices for additional evaluation ?
RESPONSE: As far as we know, the algorithms for beat-to-beat QT variability measurements have not been implemented in commercial ECG machines so far. There are, however, implementation in research version of Holter systems (e.g. by GE Healthcare). This is now shown in the manuscript. (Line 340)
Are there reference values for the normal findings ?
RESPONSE: Suggestion of normal values have been published (see our present reference [19]). However, since we show that the physiologic values are heart rate dependent, we are unable to tabulate any fixed normal values – distribution of the measured values are shown in Figure 1 from which the spread of normal limits could be derived. We have now included this possibility in the Discussion section. (Lines 355-358)
Round 2
Reviewer 1 Report
I have no further queries.